# Stem Cells in Autologous Microfragmented Adipose Tissue: Current Perspectives in Osteoarthritis Disease

**DOI:** 10.3390/ijms221910197

**Published:** 2021-09-22

**Authors:** Francesco De Francesco, Pasquale Gravina, Alice Busato, Luca Farinelli, Carlo Soranzo, Luis Vidal, Nicola Zingaretti, Barbara Zavan, Andrea Sbarbati, Michele Riccio, Antonio Gigante

**Affiliations:** 1Department of Plastic and Reconstructive Surgery-Hand Surgery Unit, Azienda ‘Ospedali Riuniti’, 60126 Ancona, Italy; francesco.defrancesco@ospedaliriuniti.marche.it (F.D.F.); michele.riccio@ospedaliriuniti.marche.it (M.R.); 2Clinical Orthopaedics, Department of Clinical and Molecular Science, Polytechnic University of Marche, 60126 Ancona, Italy; pasquale.gravina93@gmail.com (P.G.); farinelli.luca92@gmail.com (L.F.); 3Department of Neuroscience, Biomedicine and Movement, Human Anatomy and Histology Section, University of Verona, 37135 Verona, Italy; alice.busato@univr.it (A.B.); andrea.sbarbati@univr.it (A.S.); 4Fidia Advanced Biopolymers, 35031 Abano Terme, Italy; carlo.soranzo@fidiapharma.it (C.S.); luis.vidal@fidiapharma.com (L.V.); 5Clinic of Plastic and Reconstructive Surgery, Academic Hospital of Udine, Department of Medical Area (DAME), University of Udine, 33100 Udine, Italy; zingarettin@gmail.com; 6Department of Translational Medicine, University of Ferrara, 44121 Ferrara, Italy

**Keywords:** osteoarthritis, stromal vascular fraction, mechanical disaggregation, regenerative surgery, adipo micrografts

## Abstract

Osteoarthritis (OA) is a chronic debilitating disorder causing pain and gradual degeneration of weight-bearing joints with detrimental effects on cartilage volume as well as cartilage damage, generating inflammation in the joint structure. The etiology of OA is multifactorial. Currently, therapies are mainly addressing the physical and occupational aspects of osteoarthritis using pharmacologic pain treatment and/or surgery to manage the symptomatology of the disease with no specific regard to disease progression or prevention. Herein, we highlight alternative therapeutics for OA specifically considering innovative and encouraging translational methods with the use of adipose mesenchymal stem cells.

## 1. Introduction

Osteoarthritis (OA) is a chronic disease caused by biomechanical deterioration of the joint, specifically cell stress and extracellular matrix degeneration due to injury and dysfunctional repair mechanisms involving the activation of pro-inflammatory processes of innate immunity [1]. Approximately 10% of men and 18% of women above 65 years of age present symptomatic OA, and 250 million people worldwide are affected by asymptomatic OA [2,3]. Risk factors can be genetic and non-genetic; the first groups include obesity [4], age, previous traumas, bone malalignment or joint instability [5], together with other risk factors including sedentary lifestyle, postural defect. Genetical problems are involved in the balance between catabolic and anabolic activity of intra articular cartilage because of alteration of signaling pathways regarding Trasforming Growth Factor-beta/small Mother Against decapentaplegic (TGF-B/Smad), Wingless Int (Wnt/B-catenin) and Indian hedgehog/parathyroid hormone-related protein (PTHrP). Unlike other inflammatory diseases (such as rheumatoid arthritis), in which the pathogenesis has been clarified, and therefore the related therapies have been validated, the pathophysiology of osteoarthritis is still unknown.

Numerous recommendation for osteoarthritis diagnosis have been developed, such as biomarkers to expect both biological activities as natural disease progression, although given the importance of therapeutic monitoring, these biomarkers do not have a significative impact on diagnosis. The actual main purpose is to develop systems useful for quantification of flogistic assessment of OA through to the whole inflammatory markers such as cytokines, chemokines, collagen proteins, mainly in the so-called “early OA”, which is characterized by clinical and radiographic signs lack. Some authors have evidenced emergent platforms and technology that can be used in diagnostic routine both for monitoring disease progression and for better assessment of OA. Mobasheri and colleagues evidenced the main technologies and their applications through the analysis of multiplexing inflammatory biochemical markers for an efficient characterization of OA [6]. Current treatment is not effective. Mechanical stress is a result of these triggering factors, thus leading to a gradual degradation of protective cartilage between the joints, which will subsequently provoke chronic pain and further impairment. OA, besides, constitutes a substantial burden on the annual economy. Pain is a fundamental issue in OA, but pain therapy and changes in lifestyle are unsatisfactory management options, making OA a challenging disease to treat [7]. To date, quality of life for symptomatic patients may be improved solely by surgical intervention to replace joints during the final phase of disease [8]. Prevention of the progression of OA still remains an enigma, but novel biological therapies have been investigated in an effort to decelerate disease development during the early phase of asymptomatic OA towards a change in the evolution of the disease. Studies have assessed the efficacy of novel treatment options such as intra-articular applications of corticosteroid injections, hyaluronic acid injections, platelet-rich plasma (PRP) or autologous micro-fragmented adipose tissue with stromal vascular fraction for secondary prevention [9,10,11].

We assessed recent studies regarding OA pathophysiology and current treatment procedures with a view to demonstrating the role of autologous micro-fragmented adipose tissue with stromal vascular fraction to secure cartilage tissue of the joint in patients affected by OA.

## 2. Pathogenesis and Histology of Osteoarthritis Disease

Osteoarthritis was first described as a cartilage disease and subsequently a subchondral bone disorder until recent studies observed OA as an entire joint disorder involving all the surrounding tissues of a joint (articular cartilage, subchondral bone, synovium, infrapatellar fat pad, ligaments and tendons).

### 2.1. Articular Cartilage (AC)

The Articular Cartilage is characterized by avascular, alymphatic, and aneural tissue with a sole cell type of hyaline-type chondrocytes [12]. The chondrocytes are located within the extracellular matrix (ECM) and housed in the “lacunae” of the ECM or organized in small single-cell groups which derive from named “isogenic groups” (Figure 1A). ECM (or intercellular substance) is amorphous with abundant aggrecan, collagen fibers (types III, VI, IX, XI), proteoglycans (decorin, byglican and fibromodulin) and glycosaminoglycans but mainly lacking in elastic fibers [13]. AC are arranged in four layers (Figure 1B,C): 1.Tangential zone (superficial with disk-shaped chondrocytes and horizontally located collagen fibrils; 2.Transitional zone (middle): Collagen fibrils in the middle zone are diagonally oriented and round chondrocytes are scattered irregularly; 3. Radial Zone (deeper): chondrocytes are vertically positioned with radially arrayed collagen fibrils; 4. High-mineralized zone of calcified cartilage in which the collagen fibrils are perpendicular to the articular surface.

The role of AC in the development of degenerative osteoarthritis occurs via mechanical and subsequent biological mechanisms and is activated through mechanical loading and specifically via mechanoreceptors—mechanosensitive ion channels and integrins [14,15] found on the surface of chondrocytes. Histologically, the first sign of OA is a superficial cartilage fibrillation in the superficial zone. In this early phase of OA, metalloproteinases play a key role in the matrix degradation process. Metalloprotein-1 and -3 (MMP-1 and MMP-3) are responsible for the destruction of the collagen network and, in particular, MMP-13 stimulates the degradation of type II collagen, which is the most expressed in the hyaline cartilage of the joints [16]. As the matrix degeneration progresses, the chondrocytes, on the one hand, initiate a hypertrophic process [17], due to self-production of the matrix constituents and, on the other hand, the products of cartilage degradation trigger the production of proinflammatory mediators. This phenomenon is histologically visible considering the fibrillations that extend deeply into the transitional and radial zones, forming deep fissures affecting the calcified cartilage and the subchondral bone.

### 2.2. Subchondral Bone (SB)

Subchondral bone is composed of a plate-like cortical bone layer below the calcified cartilage (subchondral bone plate), and a deeper subchondral trabecular or spongy bone layer [18]. The structure (Figure 1D,E) is related to the activities of the cell population made of osteoblasts and osteoclasts. In established OA, the SB undergoes deterioration phenomena at an early stage [19] and sclerotic phenomena at a later stage [20]. These changes are accompanied by alteration in volume and thickness of the subchondral trabecular plate, leading to an osteoblastic/osteoclastic imbalance. This alteration results in micro-fractures of the SB, subchondral bone cysts, and formations of osteophytes [21]. Micro-fractures of SB, which are generally associated with cartilage erosions and fissures, are responsible for the first clinical manifestations of OA, such as pain. This phenomenon is due to the infiltration, through the fissures (bone and cartilage) of newly formed vessels and the nerves that uncommonly access certain anatomical structures. Clinically, this pathological manifestation is correlated with subchondral sclerosis, which is synonymous with late-stage OA.

### 2.3. Synovium and Synovial Fluid

The synovium is composed of the synovial membrane and synovial fluid. The synovial membrane contains some cell layers rich in fibroblast-like synoviocytes, which form a layer covering the joint cavity and the synovial liquid. Synoviocytes mainly produce lubricating molecules such as hyaluronic acid or nutrients derived from plasma, while the synovial fluid is important in avascular cartilage nutrition. The synovial membrane (Figure 1F) includes a lining or intima layer and a sub-lining or sub-intima layer that, unlike other epithelial membranes, are not separated by a basement membrane [22]. The lack of basement membrane is responsible for the incursion of serum components from the capillaries into the synovial space. The only adhesion molecules present are cadherin 11 [23], which is largely responsible for adherence between the individual synoviocytes. Histologically, the two layers may be disjointed, but the synoviocytes show a high degree of heterogeneity. The fibroblast-like synoviocytes are type-B synoviocytes that differ from type-A synoviocytes that are indeed macrophage-like synovial cells. Macrophage-like synoviocytes are CD163- and CD68-positive, which proliferate during inflammation. Fibroblast-like synoviocytes are CD55-positive and mainly produce hyaluronic acid. These differences seem to be crucial to the specific tissue reaction occurring under disease conditions such as OA [24]. The inflammation induced by OA causes synovial proliferation that provokes pervasion of T-lymphocytes, B-lymphocytes and mast cells [25,26].

### 2.4. Intra-Articular Adipose Tissue Structure

Intra-articular adipose tissue is also known as articular fat pad (AFP) and defined within the synovial joint (Figure 1F). Intra-articular fat is a fundamental part of the joint support structures, which contribute to stability and shock absorption of the joint. The Hoffa’s fat pad is an articular fat depot of the knee joint and possibly found in the hip joint [27] in proximity of the olecranon, coronoid and the radial fosse, as well as in the radio-humeral joint, the lumbar facet joints [28] and the inter-metacarpal joint [29]. AFPs are similar in structure to subcutaneous white adipose tissue (WAT), differing in the regulation of independent metabolic circuits. Adipose tissue may be considered an organ [30] owing to its ability in secreting cytokines, interleukins, growth factors and adipokines. Such growth factors are present in synovial fluid [31] and can determine the metabolism of cartilage and synovium [32]. In particular, leptin and adiponectin are adipokines secreted in large quantities only by adipose tissue, but found also in the synovial fluid. Leptin is present in greater quantities than adiponectin, and both are able to stimulate the production of pro-inflammatory mediators [33]. Leptin is able to stimulate IL1beta production and initiate MMP expression in OA cartilage. Moreover, adiponectin is able to induce MMP1 and IL6 production in synovial fibroblasts. The presence of these adipokines is associated with a high permeability of the inflamed synovium [34].

### 2.5. Osteoarthritis: Cartilage Damage, Repair and Regeneration

The OARSI (Osteoarthritis Research Society International) (Figure 2) Assessment System considers histologic features of OA (Table 1). OA severity may be classified into six grades, with Grade 0 as the value revealing normal cartilage, Grades 1–4 accounting for articular cartilage damage only, and Grades 5 and 6 including the subchondral bone [35] (Figure 2). The histologic features of OA grading classification involve the vertical depth, and the score is progressive with depth. Moreover, with the progression of disease, osteoarthritic changes are also observed in the adjacent parts of cartilage with subsequent involvement of the complete joint area. The histologic features of OA staging classification are the surface, area or volume extent, and the score progressing with length or volume. The disease involves degeneration of articular cartilage, low-grade synovial inflammation, and modifications in the joint soft tissues and subchondral bone [36].

Inflammation is an end product of these developments and is the main component of OA where the synovium produces catabolic and pro-inflammatory mediators such as cytokines, chemokines and adipokines, generating also nitric oxides, prostaglandins E2 and neuropeptides. These factors cause instability between degradation and repair of the cartilage matrix. These alterations worsen the synovial inflammation further, producing a vicious circle that exacerbates the symptoms and degeneration of the joint [37].

Fundamental repair processes of the damaged cartilage are greatly limited by the characteristics of the tissue possessing scarce potential to self-repair. Relocation to the site containing impaired chondrocytes, macrophages and blood cells is feasible only if there is interruption of the underlying subchondral bone. This repair process results in the formation of a fibrin clot. Therefore, resident stem cells guide the matrix repair process replacing the fibrin clot, differentiating into new chondrocytes, and secreting a proteoglycan-rich matrix to reshape the defective site [38], which will result in the development of weak fibrous tissue, yielding, once more, deterioration of cartilage and recurrent complications. Ineffective repair is due both to a shortage of blood vessels, which are critical for an efficient response, but also to a low presence of chondrocytes that are unable to migrate, from the lacunae, to the damaged area. Therefore, an extrinsic intervention is often necessary to regenerate impaired tissue and above all to deter disease progression. The most effective treatment, in a later phase of the disease, is replacement of the joint with a prosthesis, but many other surgical treatments have been proposed instead of joint replacement, such as osteotomy and arthroscopy [39]. The OA has been followed up, by time, with plain X-ray radiography, an exam to evaluate the treatment. Anyway, the cartilage thickness remains a key parameter to determinate the efficacy of new treatment. We can thus assess that treatment has to consider OA pathogenesis, in fact, recent emerging therapies include mesenchymal stem cells (MSCs), growth differentiation factor 5 (GDF5), platelet-rich plasma (PRP), fibroblast growth factor 18 (FGF-18), bone morphogenic protein 7 (BMP-7), WNT signaling pathway inhibitors, disintegrin and metalloproteinase with thrombospondin motifs (ADAMTS) inhibitors, and matrix metalloproteinase (MMP) inhibitors [40].

## 3. Regenerative Treatment for Osteoarthritis Disease

OA is an idiopathic disorder, and the management reflects the lack of understanding of the disease. The non-surgical approach involves the usage of treatment such as physiotherapy, kinesitherapy, weight control, drugs. The main drugs used are hormones (parathyroid hormones, calcitonin, leptin), which act as regulating molecular pathways of cartilage metabolism; bisphosphonates (zoledronic acid, alendronate) which act as improving bone metabolism, reducing bone reabsorption; monoclonal antibodies (bevacizumab, adalimumab,) which act on articular cartilage promoting collagen production; statins (atorvastatin, which reduces cartilage degradation); supplements (glucosamine, chondroitin sulfate, vitamin C, vitamin D, Selenium, Zinc, Magnesium), which act as a cartilage nourishment [41,42]. In view of this, research has been conducted on regenerative alternatives to treat OA in an attempt to ease pain and diminish symptoms by regenerating tissue and maintaining stability in local cells. Repairing articular cartilage is a challenging enterprise due to poor vascularization and innervation of articular cartilage that does not allow for the production of adequate pro-inflammatory inhibitors. The purpose of these innovative therapies, thus, is to stimulate local tissues in the production of mediators that are able to reverse the degenerative process, especially in the initial stages of OA.

### 3.1. Platelet-Rich Plasma (PRP)

Platelet-Rich Plasma (PRP) may be described as plasma volume platelet concentration derived from centrifuged whole blood [43]. Numerous platelets or thrombocyte functions involve damage to tissue, and platelet activity leads to the release of proteins and molecules that are related to vasoconstriction, inflammation, immune reaction, angiogenesis and the repair of tissue. PRP is rich in growth factors such as platelet-derived growth factor (PDGF), vascular endothelial growth factor (VEGF), transforming growth factor beta (TGF-b), epidermal growth factor (EGF), fibroblast growth factor (FGF) and insulin-like growth factor (IGF) [44]. Various preparations have been obtained [45], with or without leukocytes [46,47]. Some authors have highlighted that, in the case of intra-articular infiltration, the presence of leukocytes is of vital importance as they produce metalloproteinases and cytokines that are able to reduce inflammation and pain [48,49] as well as release mediators that trigger a cartilage repair process [50]. In contrast with the evidence reported by “in vitro” studies, where a cellular pro-inflammatory response appears to be induced by the presence of leukocytes, other authors suggest that the presence of leukocyte-rich PRP does not induce a relevant in vivo upregulation of pro-inflammatory mediators [51]. PRP is efficient in mediating fundamental elements such as chemokines, cytokines, growth factors, adhesive proteins, proteases and other small molecules (ADP, Serotonin, Calcium, Histamine and Epinephrine). In addition, safety of PRP has been observed in repeated administration of intra-articular PRP to manage moderate pain, swelling and effusion [52]. Furthermore, investigations report beneficial outcomes of PRP to reduce joint pain in the knee affected by OA in a period lasting from 6 to 12 months [53]. Novel trends are considering the application of PRP intraosseously [54]. Besides, an observational study has recently reported improved results at 6 and 12 months on intraosseous and intra-articular application of PRP compared to the intra-articular administration alone [55]. However, evidence of overall benefits is still low and is most likely due to scarce standardization of platelet-rich-plasma therapeutics.

### 3.2. Mesenchymal Stem Cells Therapy

Different approaches have been performed as potential regenerative solution for osteochondral replacement: osteochondral autografts and allografts or autologous chondrocyte implantation [56]. Moreover, the clinical use of these techniques is limited by tissue availability, donor site morbidity and unsuccessful integration. In response to limitation with the use of cells in the osteochondral grafts, mesenchymal stem cells (MSCs) have been identified in the field of regenerative surgery. Mesenchymal stem cells (MSCs) are located in numerous tissues and may be defined as specialized precursor cells. MSCs are able to self-generate and, via relevant signals, may differentiate into different tissue-specific adult cells. In that way, MSCs will substitute aged or impaired cells [57]. MSCs form the tissues of the mesodermal line such as cartilage, bone and adipose tissue, as well as tissue such as the intervertebral disc, ligaments and muscles [58]. The International Society of Cellular Therapy [59] established a set of defining characteristics for MSCs which include the ability to adhere to plastic, expression of surface markers CD73, CD90, CD105 and a lack of hematopoietic markers CD34, Cd45, CD14, CD19. Moreover, characteristics would include tripotent differentiation into chondrogenic, osteogenic, and adipogenic phenotypes. In addition to their differentiating capacities, MSCs also represent noteworthy potential in regenerative medicine due to their anti-inflammatory and immunomodulatory potential [60]. MSCs are considered fundamental in tissue engineering since they are able to differentiate into terminal specialized cells, although currently, MSCs are exploited to “convince” the tissue or organ to self-regenerate. Regenerative medicine intends restoration principally via cell provision and specific stem cells that further enhance regeneration. It is therefore valid to define regenerative medicine as the restoration of human cells, tissue or organs to maintain regular functionality [61]. As aforementioned, the beneficial effects of MSCs are due to enhancement of both viability and proliferation of native cells, mitigation of cell death, delay cell senescence and anti-inflammatory and immunomodulatory effects. These reparative actions are obtained through MSC-secreting paracrine growth factors and cytokines, dynamic and direct cellular inter-communication along with extracellular vesicle release (defined exosomes)-containing peptides, mRNA and microRNAs [62,63,64]. The regulation of stem cell renewal and differentiation occur in the “niche” [65]. Multiple niches may be observed in different tissues [66] and stem cells contained here within have been utilized to repair cartilage. MSCs have been found in bone marrow [67], adipose tissue [68], dental pulp [69], umbilical cord tissue [70], but also in resident joint tissue such as the articular cartilage, synovium, periosteum, infrapatellar fat pad and trabecular bone [38]. The efficacy of stem cell treatment for OA has not yet been defined, but the secretion of anti-scarring (KGF, SDF1, MIP1a, MIP1b), anti-apoptotic (STC-1, SFRP2, TGFbeta1, HGF), angiogenic (VEGF), and mitogenic (TGF-a, TGF-b, HGF, IGF-1, FGF-2, EGF) factors may explain the natural repair mechanisms [71]. Other investigations have demonstrated a possible interaction between immune cells and MSCs as well as the potential to restrain propagation of inflammatory T cells and development of monocytes and the ability to impede B cell activity, which interfere with the underlying pathological or inflammatory process [72]. AI Caplan has recently proposed that the pericyte is released from its position in the vascular network in the case of a focal injury and have an immunomodulatory function. This immune modulation turns off T-cell surveillance of the injured tissue and thus provides a blockage of immune responses, while its trophic activity ensures that the field of damage is limited, that scarring does not occur and that tissue-intrinsic progenitors replace the expired cells. Angiogenesis occurs via MSC secretion of bioactive factors, such as VEGF, and via stabilization of newly forming vessels [73]. The new era of cell-mediated therapy (Figure 3) in the clinical trial database is promising. Multiple ongoing trials involving MSCs are evidence of the growing interest and viability of these cells. Further investigations are required to assess safety and subsequent efficacy with an urgency of broad diffusion of publications within the scientific communities to better understand therapeutic options.

### 3.3. Intra-articular Application of Autologous Microfragmented Adipose Tissue with Stromal Vascular Fraction

Adipose stem cells (ASCs) are able to renew themselves and create multiple lineages [74]; besides, they can readily and rapidly expand in vitro, will not age easily and provoke fewer morbidities in patients [75]. ASCs have also displayed significant potential to propagate and differentiate into mesoderm-like tissue in relation to bone marrow derived- MSCs or other sources [76]. Importantly, ASCs are easily isolated and particularly accessible from subcutaneous adipose tissue [77,78,79]. ASCs have also shown efficient chondrogenic differentiation during in vitro expansion under adequate conditions [80]. In any case, physical and mechanical factors are required to perform an adequate formation of the cartilage tissue in vivo such as mechanical stimuli or a particular texture of the scaffolds [81,82]. The clinical use of ASCs is strictly regulated, because these products are considered “drugs” and therefore particularly restricted in clinical practice in Europe and the USA [83]. Such restrictions have led to novel studies regarding ASC alternative therapeutics considering “minimal manipulation” [84]. In particular, if ASCs are not expanded in vitro but extracted from the adipose tissue within the operating room without substantial manipulation and without use of collagenase, then the United States Food and Drug Administration (US FDA)/European Medicine Agency (EMA) allow such treatments [85]. Enzymatic tissue digestions are considered by the FDA (and EMA) as “substantial manipulations” and have accordingly imposed important restrictions. The issue of ASC “minimal manipulation” is considered during the isolation of several cells’ populations using mechanical processes to adhere to the regulations set by the FDA and EMA worldwide [86]. Additionally, alteration of biological, physiological, or structural features of cells or tissues is considered as important manipulation. The bone marrow aspirate concentration is an invasive procedure provoking donor morbidity, while the liposuction for obtaining SVF is a minimally invasive procedure [87]. Albeit efficient, the enzymatic digestion necessitates xenogenic substances that may cause immune reactions and is discordant with the European Good Manufacturing Practice (eGMP) Guidelines (Regulation (EC) No. 1394/2007 of the European Parliament and the European Council). To elude this problem, single devices have been adopted to separate and isolate SVF from adipose tissue [88,89]. Non-enzymatic methods to isolate SVF use mechanical or physical forces to manipulate the structural integrity of adipose tissue. These procedures are less specific and are sufficiently able to displace SVF cells from their own niche, and some authors have consequently introduced the concept of a stromal vascular niche [90,91]. The end product acquired via non-enzymatic digestion is not strictly cellular stromal vascular material, as would be generally acquired via enzymatic digestion, but a combination of cellular debris, blood cells, and components of ECM [92]. Moreover, the mechanical devices can preserve cells in clusters, or rather, in their native environment, which will aid in retaining cell function, including exosome discharge and secretion. The stromal vascular niche, therefore, protects the activated ASCs, enhancing their potency in the recipient environment, but also actuates a cascade of biological events that mimic the natural healing process. Non-enzymatic procedures have been proposed including mechanical dissociation of adipose tissue using closing devices and operator-dependent tools (Figure 4).

These devices differ from each other in the isolation protocol, in time and in the category of tissue dissociation, but also vary in the final SVF product. Non-enzymatic isolation methods are based on centrifugation force, pressure, filtration and washing. Mechanical systems commonly used to harvest and purify adipose tissue to obtain SVF are: Puregraft (Bimini Technologies LLC, Plano, Texas, USA), LipiVage (Genesis-Byosystems-Inc, Lewisville, Texas, USA), Lipogems (Lipogems Int Spa, Milan, Italy), Rigenera (HBW srl, Turin, Italy), Lipo-Kit GT (Medikan-International Inc, Seul, Korea), Hy-Tissue Nanofat (Fidia Farmaceutici, Abano Terme, Italy), Hy-Tissue SVF (Fidia Farmaceutici, Abano Terme, Italy), StromaCell (Micro-Aire-Surgical Instruments, Charlottesville, Virginia, USA), MyStem (MyStem LLC, Wilmington, NC, USA), Revolve (Life Cell Corporation, Branchburg, New Jersey, USA), Wal Body-Jet and Q-Graft system (Human Med AG, Schwerin, Germany), IntelliCell (Biosciences Inc, New York, NY, USA). Many of the devices reported have received evaluations in pre-clinical and clinical trials. Older systems are LipiVage and PureGraft, which were among the first products to be commercialized [93,94]. The LipiVage collection, washing and transfer technology is a device that allows collection of adipose grafts in controlled conditions with low vacuum, avoiding centrifugation or decantation. The lipoaspirate fat, inside the cannula, is separated from oils and fluids by an integrated filter in an extremely short time (15 min). In addition, fragmented adipose tissue from LipiVage showed no differences by normal adipose tissue, yielding large-sized grafts. However, an analysis of particles has not yet been conducted. The PureGraft technology is based on the filtration of adipose tissue through a particular membrane, an equally rapid procedure (15 min). In addition, grafts from PureGraft displayed larger particles (>1000 μm) and were able to operate a “dialysis” of the adipose tissue without resorting to other, more destructive methods such as centrifugation [95]. The main use of these technologies is in the field of fat grafting for breast volumes [96]. The most studied and commonly used system in clinical practice is the LIPOGEMS device. This technology is a closed device that allows collection of uniform products containing pericytes/ASCs with a slight mechanical force The end product is adipose tissue reduced into small fragments (600 / 400 μm), which progressively reduce in size and are without oil or blood residues, rich ASCs [97,98]. This device has been widely used especially in orthopedics for the treatment of tendinopathies and osteoarthritis [99,100]. Moreover, some authors have devised a “pure” system of mechanical disintegration [101] of tissues that is easy to use, less expensive and faster. This technology, called Rigenera micrografting technology, can disaggregate autologous tissue, with a calibrated size of 80 mm, collecting autologous micrografts enriched in progenitor cells, growth factors, and particles of ECM, by in vitro studies [84,102]. Some authors have performed comparative analyses between different mechanical and enzymatic systems. Raposio et al. [103,104] compared two procedures for isolation of ASCs, based on enzymatic + mechanical (centrifugation/vibrating plus collagenase) and mechanical (centrifugation or vibrating) methods. The authors showed that the enzymatic + mechanical procedure endorsed a major number of ASCs compared to the mechanical method alone. Indeed, Domenis et al. [105] showed that ASCs obtained from a mechanical device (Fastem kit) was less efficient in relation to the enzymatic tools (Lipo-kit and Celution). All three procedures, nevertheless, were able to maintain the amount of adipose tissue and thickness in the reconstructed breast. Additionally, Senesi et al. [106] showed good cell viability, CD markers expression, and differentiation potency of ASCs obtained from mechanical devices (Rigenera and Lipogems) compared to enzymatic digestion. Furthermore, the authors asserted that the mechanical methods acted differently on the release of the ASCs from the SVF perivascular niches. Only enzymatic digestion was able to acquire a “pure” cell population and ASCs could rapidly differentiate into all mesodermal lines. Of the two mechanical systems analyzed, only the micro-grafts obtained by Rigenera (compared to Lipogems) were able to differentiate into all mesodermal lines, albeit more slowly than by enzymatic digestion. Recently, some authors have studied a new promising device (Hy-Tissue SVF) that allow the stromal vascular fraction to isolate in the form of free cells and micro-fragments (30 / 70 μm) of connective tissue containing stromal cells and extracellular matrix [107]. This system is able to disaggregate autologous adipose tissue using a double bag with an inner filter bag of 120 μm mesh by using a small plastic rod. The main structural and morphological unit, the adipose niche, is maintained after disintegration and protects the activated ASCs, strengthening their effectiveness in the receiving environment. This is the main difference between this system and the others, because the preservation of the adipose structural niches increases the effectiveness of the ASCs. In addition, the elimination of enzymatic action will reduce tissue trauma while maintaining cellular integrity. The reduction in the size of adipose clusters favors engraftment because of a more convenient, more effective, and rapid revascularization of the micrograft owing to the interaction with the receiving vascular microenvironment.

### 3.4. Exosome and Extracellular Vescicles (EVs)

Recently, studies have revealed that MSCs are able to modulate the gene expression of the surrounding cells through miRNA secretion and provide relevant exosome involvement in the benefits of MSC-based therapy [108]. Exosomes are extracellular vesicles with a diameter measurement ranging from 30 to 150 nm. In the course of multi-vesicular body development, inward budding of endosomal membranes is observed which contributes to the fundamental inter-cell communication. The multi-vesicular body endosomes fuse with the cell membrane leading to secretion of exosomes [109,110]. Most of the MSC paracrine factors are crucial to tissue regeneration and lined to the discharge of EVs. MSC exosomes (Figure 5) originate in adipose tissue, the bone marrow and other tissues and bear a rich and complex load of nucleic acid (mRNA and miRNA), proteins and lipids [111].

In OA, the in vitro studies revealed chondroprotective and anti-inflammatory functions of exosomes, as observed in chondrocyte models [112]. Moreover, in various studies, the significance of exosomes has been demonstrated as regards the benefits of MSC-based therapies in treating cartilage lesions and OA. Recent outcomes in pre-clinical trials have shown efficacy of MSC exosomes in cartilage repair and renewal, enhancing chondrocytes to amalgamate type II collagen and reduce production and expression of ADAMTS-5. Such a development will ensure ECM [113] as well as boost restoration of cartilage via paracrine signaling mechanisms along with secretion of soluble trophic factors [114]; reduction in inflammatory markers (iNOS) [115] and downregulation of inflammatory signals by secretion of IL-1, IL-6, IL8, MMP-1 and MMP-13 [116]. EVs that are isolated from human ASCs use various chondroprotective mechanisms to reduce inflammatory mediators (TNF-alpha, IL-6, PGE2, NO) and minimize MMP activity therefore enhancing generation of the anti-inflammatory cytokine IL-10 [117]. Many authors demonstrated, also, that exosomes from ASCs could inhibit and defer cartilage deterioration in OA models through the suppression of catabolic molecules [118] and through the immunoregulatory stimuli of hyaluronan [119]. These pioneering findings have consolidated the positive outcomes of ASC-derived EVs as a new therapeutic alternative for OA. Adequate investigations are scarce, especially considering these therapeutic options as authentic products of mechanical digestion activated by the various available medical devices. In any case, a detailed analysis to investigate the functions and mechanisms of exosomes in clinical practice is urgently required taking into account the positive outcomes of preclinical studies. In clinical trials, therapy based on exosomes derived from ASCs should aim to optimize criteria involving exosome concentration and dosage, with injection times and intervals. Additionally, the immune response in individuals is to be assessed following exosome administration.

## 4. Future Perspectives

Subcutaneous adipose tissue has aroused remarkable interest in the field of plastic surgery and regenerative medicine within the last decade with successful use of SVFs and ASCs observed in clinical studies. Nevertheless, limitations are still evident regarding different therapeutic procedures and above all with the diverse regulations in European and non-European countries. In particular, some non-European countries continue to use collagenase. For this reason, the International Federation for Adipose Therapeutics and Science (IFATS) is developing a common standard operating protocol using toxin-free and xenofree products [120]. Consequently, attention is increasingly turning to closed non-enzymatic systems as they represent the safest and most effective means for obtaining stromal vascular fraction from adipose tissue. Obviously, each procedure is characterized by distinct advantages and disadvantages. Therefore, continuous development and optimization for obtaining SVF is essential, especially for the purpose of critical parameters such as washing, the filtration system, centrifugation and size of the cannulae. These parameters require attentive analysis and comparison to produce beneficial systems and optimal “cell therapy” use. Stem cell-based therapy is of huge relevance in OA regenerative medicine. Preclinical and clinical trials have demonstrated good results in the OA treatment in particular to contribute to delay or prevent OA progression before considerable cartilage degradation. Moreover, MSC-derived exosomes have captured attention as possible therapeutic agents because they carry most of the therapeutic effect of the MSCs themselves. Exosomes may be defined as a cell-free therapy reducing safety issues concerning live-cell administration. Noteworthy is the anti-inflammatory effect of exosomes in the treatment of disorders. Furthermore, MSC-derived exosomes possess anti-inflammatory elements reaching the recipient cells and decreasing inflammation. In view of these considerations, MSC-derived exosomes may be used in various inflammatory diseases such as OA. The numerous in vitro studies yielding positive outcomes of the modulatory and protective effects on chondrocytes by exosomes produced by ASCs would elicit future studies to deepen the understanding of these medical devices. However, important factors need to be addressed before clinical application of MSC-derived exosomes. Standards for the purity of exosomes should be a priority and subsequently, quality control (QC) protocols of isolated MSC-derived exosomes need to be set. Exosomes may also be considered as vectors of potential therapeutic molecules and therefore delineate an optimal theranostic approach [3,121]. The “all-in-one approach” that characterizes the theranostic method is a promising therapeutic tool in precision medicine of OA since it permits a tailor-made approach to diagnose and monitor disease in the individual at an early phase of illness with the potential of site-specific drug delivery. Nanotherapeutics using exosomes are a ground-breaking field and is rapidly expanding to offer novel alternatives in anti-inflammatory treatment. Emerging applications that are worthy of mention are the essential therapeutic anti-inflammatory role of EVs and the natural exploitation of EVs as a carrier for small molecule drugs, therapeutic RNAs and protein delivery together with targeting moieties. Exosomes are apparently optimal vector candidates well-defined in this novel technique to identify specific disease areas producing minimal adverse responses. Last but not least, gene therapy using transfer gene and tissue engineering techniques represents a potential new strategy for the in situ treatment of osteochondral lesions. The main advantage of this therapy is represented using genic vectors through delivery system. This technology should be able to modulate the pathological process of osteoarthritis through intrinsic changes [122].A new promising therapeutic approach c is characterized by the bio fabrication of 3D structures mimicking articular cartilage propriety due to 3D bioprinting technique. This brand new technology can be used to reproduce complex scaffold characterized by cells, growth factors, extracellular matrix, to be used to physically substitute injured cartilage [123]. Some authors have biofabricated human cartilage using adipose tissue deriving from infrapatellar fat succeeding in production of hyaline cartilage on a bio scaffold in order to produce a patient tailored cartilage to replace the injured one [124]. To conclude, this technology can be used to develop in vitro model of osteoarthritis that can be used for further scientific research [125,126,127,128,129,130].

## Figures and Tables

**Figure 1 ijms-22-10197-f001:**
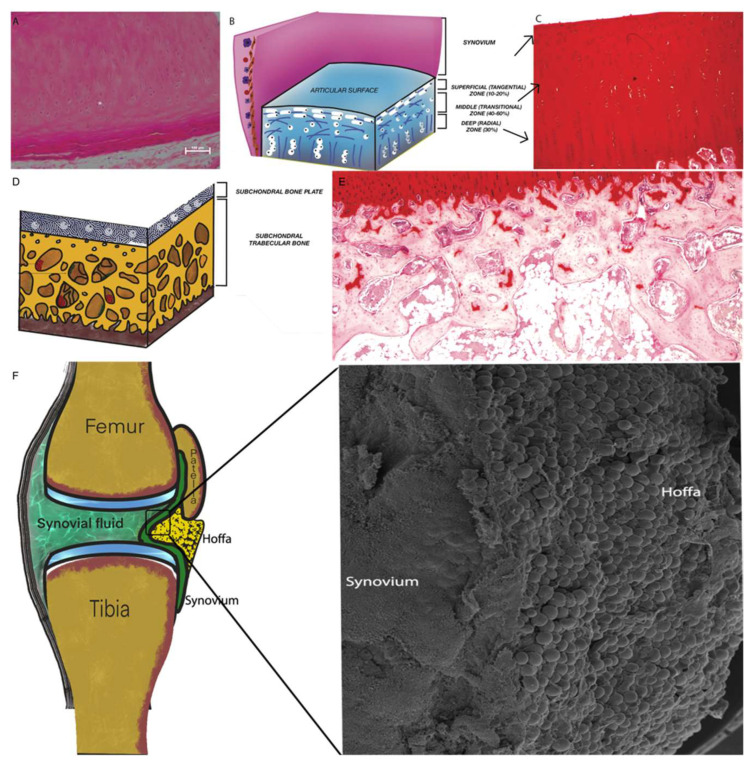
(**A**) The figure shows the histological aspect of a normal Articular Cartilage. (**B**) Schematic representation of superficial, middle and deep zones. (**C**) Histological aspect of a damaged articular cartilage. (**D**) Schematic representation of Subchondral Bone. (**E**) Histological aspect of Subchondral Bone. (**F**) The figure shows the microscopic appearance of the synovial membrane and the intra-articular Hoffa’s fat pad (Scale bar 80 um).

**Figure 2 ijms-22-10197-f002:**
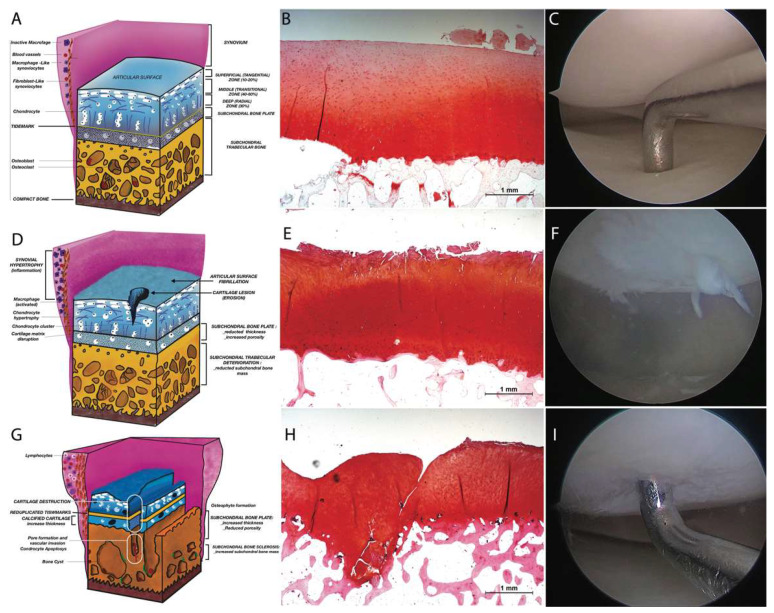
OA severity score by OARSI: (**A**): Grade 0, normal articular cartilage; (**B**): Grade 0 Histological aspect; (**C**): Grade 0 Arthroscopic aspect; (**D**): Grade 3 Vertical clefts/erosion to the calcified cartilage extending to <25% of the articular surface; (**E**):Grade 3 Histological aspect; (**F**): Grade 3 Arthroscopic aspect; (**G**): Grade 6 Vertical clefts/erosion to the calcified cartilage extending to >75% of the articular surface, (**H**): Grade 6 Histological aspect; (**I**): Grade 6 Arthroscopic aspect.

**Figure 3 ijms-22-10197-f003:**
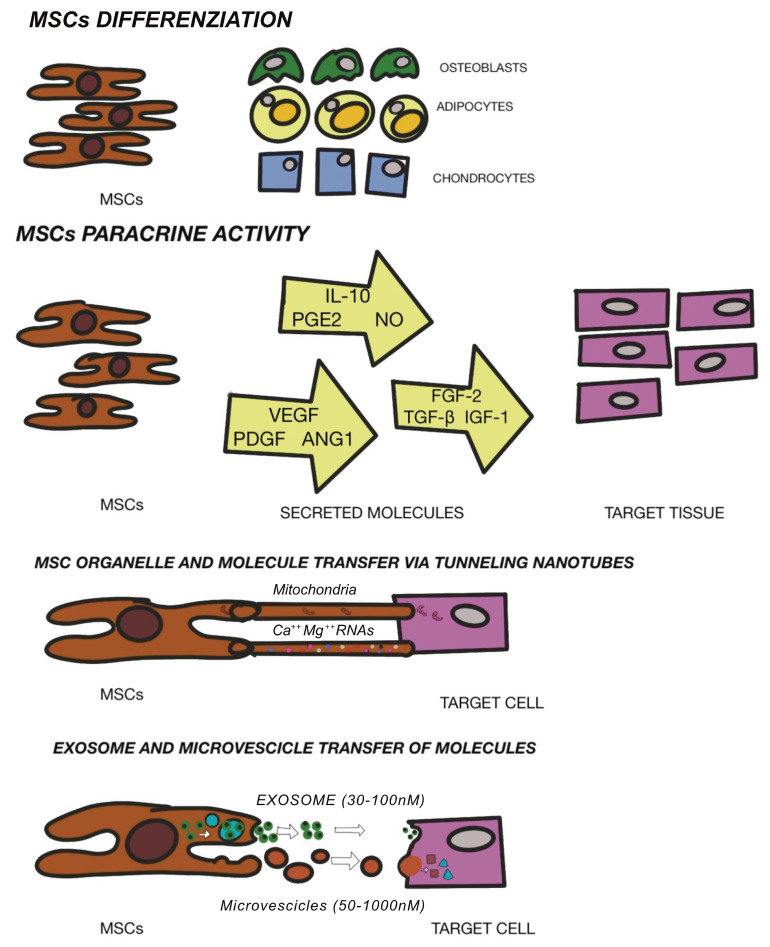
Mechanism of MSC mediated repair. Modified from Fellows CR et al. Front Genet. 2016.

**Figure 4 ijms-22-10197-f004:**
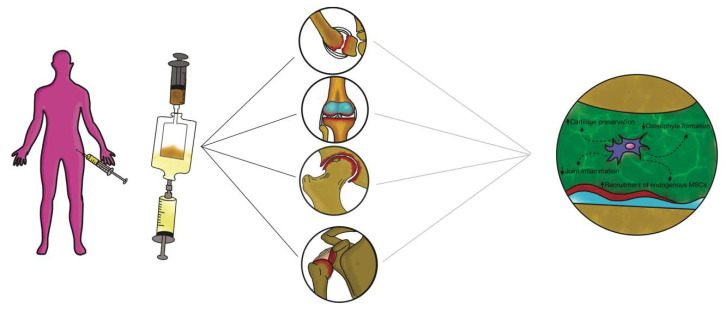
Non-enzymatic procedures have been proposed including mechanical dissociation of adipose tissue using automated closing devices and non-operator-dependent tools.

**Figure 5 ijms-22-10197-f005:**
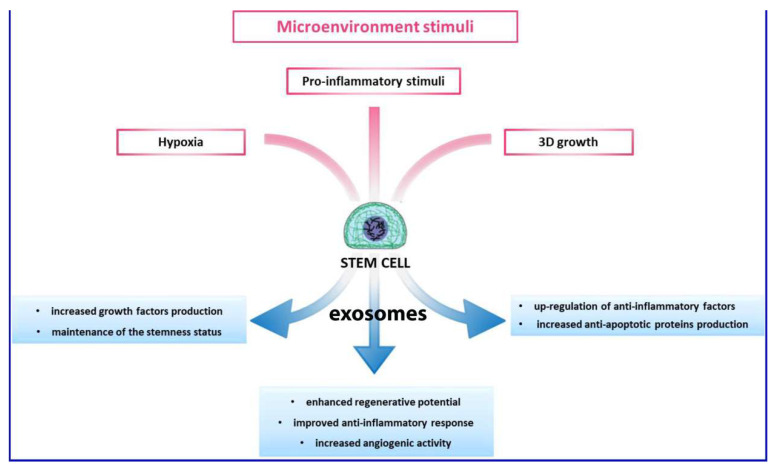
Different modes of action of exosomes.

**Table 1 ijms-22-10197-t001:** Different evaluation score for osteoarthritis.

Grade of Osteoarthritis	OARSI Score	Radiographic Score (Ahlback)	Radiographic Score (Kellgren–Lawrence)
Grade 0	Normal	No radiographic findings of OA	No radiographic findings of OA
Grade 1	Small fibrillations without loss of cartilage	Joint space narrowing <3 mm	Doubtful joint space narrowing and possible osteophytic lipping
Grade 2	Vertical clefts down to the layer immediately below the superficial layer and some loss of surface lamina	Joint space obliterated or almost obliterated	Definite osteophytes and possible joint space narrowing
Grade 3	Vertical clefts/erosion to the calcified cartilage extending to <25% of the articular surface	Minor bone attrition (<5 mm)	Multiple osteophytes, definite joint space narrowing, sclerosis, possible bony deformity
Grade 4	Vertical clefts/erosion to the calcified cartilage extending to 25–50% of the articular surface	Moderate bone attrition (5–15 mm)	Large osteophytes, marked narrowing of joint space, severe sclerosis, and definite deformity of bone ends
Grade 5	Vertical clefts/erosion to the calcified cartilage extending to 50–75% of the articular surface	Severe bone attrition (>15 mm)	
Grade 6	Vertical clefts/erosion to the calcified cartilage extending to >75% of the articular surface		

## Data Availability

The clinical data used to support the findings of this study are included within the article.

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
