# Peer review of "Stem Cells in Autologous Microfragmented Adipose Tissue: Current Perspectives in Osteoarthritis Disease"

_ijms, 2021, doi:10.3390/ijms221910197_

Round 1

Reviewer 1 Report

Minor changes to your paper:

Line 27: Write what alternatives are you proposing.

Introduction: Well written according to scientific guidelines. No further improvements.

Pathogenesis chapter: It is well written. From my perspective its a little too long. Your review focus is on stem cells in OA. I suggest you to reduce it.

Figure 1-2-3: Make a single panel. all 3 figures in 1.

Line 252: I recommend to you to write about the use of systemic drugs in OA. This is an important aspect that should be taken into account by orthopaedic surgeons.

Here are some recommendations for you: 

Systemic drugs with impact on osteoarthritis. Drug Metab Rev. 2019 Nov;51(4):498-523. doi: 10.1080/03602532.2019.1687511

Tibolone, alendronate, and simvastatin enhance implant osseointegration in a preclinical in vivo model. Clin Oral Implants Res. 2020 Jul;31(7):655-668. doi: 10.1111/clr.13602.

Line 518: Well written. I suggest to write a clear conclusion about stem cell in OA.

Author Response

Thanks to the referee.

Minor changes to your paper:

Line 27: Write what alternatives are you proposing.

thanks done

Pathogenesis chapter: It is well written. From my perspective its a little too long. Your review focus is on stem cells in OA. I suggest you to reduce it.

thanks done

Figure 1-2-3: Make a single panel. all 3 figures in 1.

thanks done

Line 252: I recommend to you to write about the use of systemic drugs in OA. This is an important aspect that should be taken into account by orthopaedic surgeons.

thanks done

Here are some recommendations for you: 

Systemic drugs with impact on osteoarthritis. Drug Metab Rev. 2019 Nov;51(4):498-523. doi: 10.1080/03602532.2019.1687511

Tibolone, alendronate, and simvastatin enhance implant osseointegration in a preclinical in vivo model. Clin Oral Implants Res. 2020 Jul;31(7):655-668. doi: 10.1111/clr.13602.

added

Line 518: Well written. I suggest to write a clear conclusion about stem cell in OA.

thanks done

Reviewer 2 Report

Authors have drafted the manuscript in a decent manner which can be followed by readers however below are few considerations to be implemented to obtain publishable standards.

  1. Introduction has to be made stronger and it is suggested to include recent technology and platforms which are being used to address the OA.
  2.  Subsection 2.0 on the pathogenesis part is really matching up with the manuscript's title and the abstract stated. It's recommended to amend and update with a focus on a therapeutic value on pathogenesis how advanced therapeutics would enhance tissue repair and regeneration.
  3. References cited by authors have to be updated with recent citations max 5 years (2016-2021).
  4. It is also recommended to add and highlight important studies performed on OA models using biofabrication techniques such as 3D printing, electrospinning and others, which would add value to the manuscript.

Author Response

  • Introduction has to be made stronger and it is suggested to include recent technology and platforms which are being used to address the OA.

thanks done

  •  Subsection 2.0 on the pathogenesis part is really matching up with the manuscript's title and the abstract stated. It's recommended to amend and update with a focus on a therapeutic value on pathogenesis how advanced therapeutics would enhance tissue repair and regeneration.

thanks done

  • References cited by authors have to be updated with recent citations max 5 years (2016-2021).

thanks done

  • It is also recommended to add and highlight important studies performed on OA models using biofabrication techniques such as 3D printing, electrospinning and others, which would add value to the manuscript.

thanks done

Round 2

Reviewer 2 Report

The authors have addressed all the suggested comments.